# Antidepressant Effect of Shaded White Leaf Tea Containing High Levels of Caffeine and Amino Acids

**DOI:** 10.3390/molecules25153550

**Published:** 2020-08-03

**Authors:** Keiko Unno, Daisuke Furushima, Yuzuki Nomura, Hiroshi Yamada, Kazuaki Iguchi, Kyoko Taguchi, Toshikazu Suzuki, Makoto Ozeki, Yoriyuki Nakamura

**Affiliations:** 1Tea Science Center, Graduate Division of Nutritional and Environmental Sciences, University of Shizuoka, Shizuoka 422-8526, Japan; gp1719@u-shizuoka-ken.ac.jp (K.T.); yori.naka222@u-shizuoka-ken.ac.jp (Y.N.); 2Department of Neurophysiology, School of Pharmaceutical Sciences, University of Shizuoka, Shizuoka 422-8526, Japan; iguchi@u-shizuoka-ken.ac.jp; 3Department of Drug Evaluation & Informatics, School of Pharmaceutical Sciences, University of Shizuoka, Shizuoka 422-8526, Japan; dfuru@u-shizuoka-ken.ac.jp (D.F.); m15088@u-shizuoka-ken.ac.jp (Y.N.); hyamada@u-shizuoka-ken.ac.jp (H.Y.); 4Tea Research Center, Shizuoka Prefectural Research Institute of Agriculture and Forestry, Kikugawa, Shizuoka 439-0002, Japan; toshikazu1_suzuki@pref.shizuoka.lg.jp; 5Nutrition Division, R&D Group, Taiyo Kagaku Co., Ltd., Yokkaichi, Mie 510-0844, Japan; mozeki@taiyokagaku.co.jp

**Keywords:** antidepressant effect, antistress effect, asparagine, aspartate, caffeine, clinical study, green tea, salivary α-amylase, theanine

## Abstract

The young leaves of green tea become lighter in color than usual when protected from sunlight by a shading net for about two weeks while growing. These leaves are called “shaded white leaf tea” or SWLT. In the eluate of SWLT, the amount of amino acids (361 mg/L) was significantly higher than that in regular tea (53.5 mg/L). Since theanine and arginine, the first and second most abundant amino acids in SWLT, have significant antistress effects, we examined the antistress effect of SWLT on humans. SWLT or placebo green tea (3 g) was eluted with room-temperature water (500 mL). Participants consumed the tea for one week prior to pharmacy practice and continued for 10 days in the practice period. The state-trait anxiety inventory, an anxiety questionnaire, tended to be scored lower in the SWLT group than the placebo, but other stress markers showed no differences. The effect of the difference in SWLT components examined with mice showed that aspartic acid and asparagine, which are abundant in SWLT, counteracted the antistress effects of theanine and arginine. Large amounts of caffeine also interfered with SWLT’s antistress effect. Thus, SWLT, which is high in caffeine and amino acids, suppressed depressant behavior in mice.

## 1. Introduction

Tea (*Camellia sinensis* (L.) Kuntze) shoots are rich in theanine, one of the umami components of tea. However, when the leaves are exposed to direct sunlight, theanine (L-theanine, Thea) is used for the synthesis of catechins and other compounds as a nitrogen source [1]. An increase in catechins and a decrease in theanine causes an increase in astringency. On the other hand, tea strains in which the green tea shoots turn yellowish-white are generally called white leaf tea. These are reported to be excellent in terms of aroma and taste according to sensory tests, because they have lower amounts of chlorophyll and catechins, and a higher amount of free amino acids, than regular green tea [2]. However, these cultivars have disadvantages, such as low yield and difficulties with propagation. Therefore, a technique has been developed in which the green tea leaves of a general cultivar, such as “Yabukita,” are turned yellowish-white by completely protecting them from light [3]. If the young leaves are protected from sunlight with a shading net for about two weeks, the decomposition of theanine is suppressed, resulting in green tea with an increased amount of amino acids compared to normal green tea [4]. Green tea prepared in this way is called “shaded white leaf tea,” or SWLT. The amino acids in shoots with 100% shading are more than doubled as compared with the levels before their covering. Theanine (Thea), which is an amino acid that accounts for about half of the total free amino acids in tea leaves, increases 1.3-fold with shading. Arginine (Arg), the second most common amino acid, increases 2.5-fold with shading. The amount of serine (Ser) and asparagine (Asn) in the tea leaves is small, but increases by 3.8- and 16.9-fold, respectively, with shading [4].

Many studies have shown that Thea exhibits an excellent antistress effect [5,6,7,8]. In addition, arginine (Arg), which is the next most abundant amino acid after Thea, also exhibits an excellent antistress effect, similar to the effect theanine has on mice [9]. Therefore, in this study we investigated the antistress effect of SWLT, which is rich in Thea and Arg, in humans.

On the other hand, tea shoots have a high amount of caffeine. Previous studies have shown that the antistress effects of Thea and Arg are counteracted by caffeine and epigallocatechin gallate (EGCG) [9]. We made a low-caffeine green tea, and examined the antistress effect on humans. The results showed that stress was reduced in participants in their 20s, 40–50s, and 90s [10,11,12]. Matcha green tea is rich in Thea, Arg, and caffeine. When the green tea ingredients were mixed with powdered feed, as in the case of matcha green tea, animal experiments clearly showed that antistress effects were not observed if the molar ratio (of caffeine and EGCG/Thea and Arg, CE/TA) was 2 or more [13]. In addition, participants in their 20s showed reduced stress with matcha with a CE/TA ratio of 1.7, but not with matcha with a ratio of 10 [13]. These data indicate that the ratio of tea components, along with Thea content, is very important.

SWLT has a high Thea and Arg content and is below a CE/TA ratio of 2, so it is expected to show an antistress effect. According to a survey by the Japanese Ministry of Health, Labor and Welfare, about 60% of men and women experience significant stress at work. Green tea is the most common beverage that people drink daily in Japan, so it would be very meaningful if it was scientifically revealed that the intake of green tea helps to maintain good mental health. Therefore, we first examined whether SWLT can reduce stress, because of the high amounts of Thea and Arg in SWLT. However, we found that the stress-reducing effect of SWLT in students was about the same as a placebo green tea. Next, we investigated the reason for this unexpected result for SWLT using an animal psychosocial stress model, and based on this data, we examined the antidepressant effect of SWLT.

Depression is one of the most common psychiatric disorders, and stress is an important risk factor for depression [14]. Tea consumption is shown to reduce the risk of depression [15], and green tea catechins have been shown to decrease depressive syndromes in experimental animals [16]. Thea has been reported to have beneficial effects on depressive syndromes, anxiety, and sleep disturbance in patients with a major depressive disorder [17]. In addition, caffeine is suggested to be a therapeutic agent for motivational dysfunction in depression [18]. Therefore, we evaluated the antidepressant effect of SWLT containing high levels of caffeine and amino acids.

## 2. Results

### 2.1. Content of Theanine, Caffeine, and Catechins in SWLT and Placebo Green Tea

The content of amino acids in SWLT was much higher than in the placebo green tea in the eluate derived from steeping for 3 h (Table 1). The placebo green tea was a medium-grade common green tea. Caffeine was about two times higher in SWLT than in the placebo green tea. Although the contents of EGCG were similar between SWLT and the placebo green tea, the content of EGC was higher in the placebo green tea than SWLT.

Since the participants ingested 500 mL of SWLT or placebo green tea, the participants that ingested SWLT consumed about 70 mg of Thea and 35 mg of Arg per day, and the participants who ingested the placebo tea consumed about 29 mg Thea and 5 mg of Arg per day. When the CE/TA ratio, which is an index of the antistress effect in the case of matcha, was also applied to the SWLT and placebo green tea, SWLT scored 1.12 and the placebo tea scored 4.47.

### 2.2. Levels of Salivary Amylase Activity (sAA), State-Trait Anxiety Inventory (STAI) Value, Physical Condition, Subjective Stress, Achievement Emotion, and Sleeping Time in Students

The level of sAA is usually low at the time of waking up but becomes high as a result of sympathetic excitement during the day [19,20]. In both groups, the level of post-practice sAA tended to be higher than the level of pre-practice sAA, but not significantly so. However, there was no significant difference in sAA levels between the SWLT and placebo tea groups in pre-practice and post-practice during routine daily life at the university and pharmacy practice (Table 2).

Next, the STAI values were examined to assess anxiety based on appraisal standards. The values were no different between the SWLT and placebo groups before the pharmacy practice. On the other hand, the values tended to be lower in the SWLT than in the placebo group after the pharmacy practice (*p* = 0.065). Physical condition was not different between the two groups during routine daily life at the university and pharmacy practice. Subjective stress was evaluated by each participant at the end of daily practice using visual analogue scales (VAS: 0–10). The average score was not different between the two groups (Table 2). Sense of achievement was evaluated by participants as an ordinal scale at the end of the daily pharmacy practice. There was no difference between the average of both groups (Table 2). The average sleeping time was not different between the SWLT and placebo groups during routine daily life at the university and pharmacy practice.

### 2.3. Antistress Effects of SWLT in a Mouse Model of Psychosocial Stress

The relationship between the intake of tea components from SWLT and the suppression of stress was examined in a mouse model. SWLT has much higher amounts of free amino acids and a different composition ratio compared with placebo green tea. When compared in molar ratios, Thea accounted for about half of the total free amino acids in placebo green tea, while it was only one-third for SWLT. On the other hand, the proportion of Arg, Gln, Asn, and Asp increased (Figure 1). Therefore, the effect of these amino acids on the stress response of mice was examined.

Adrenal hypertrophy was observed in mice stressed by confrontational housing, but not in mice that ingested Thea under stressed conditions. When the mice ingested Arg and Gln with Thea, no hypertrophy was observed, but when the mice also ingested Asn and Asp with Thea, Arg, and Gln, the adrenal gland was significantly enlarged in mice under stressed conditions (Figure 2).

Furthermore, the effects of caffeine and EGCG on Thea and Arg were also examined. Thea, Arg, caffeine and EGCG, in the same composition ratio as in SWLT or placebo tea, were added to the drinking water and allowed to be taken freely. As a result, adrenal hypertrophy was not sufficiently suppressed. The molar ratio of caffeine and EGCG to Thea and Arg (CE/TA) was 1.12 for SWLT and 4.47 for placebo tea. When caffeine and EGCG were reduced to a CE/TA ratio of 0.42, adrenal hypertrophy was significantly suppressed (Figure 3).

It has been confirmed that similar results can be obtained at different concentrations if the CE/TA ratio is the same [9]. However, it was newly revealed that Asn and Asp act antagonistically to the antistress effect of Thea (Figure 2). Therefore, to clarify the role of Asp in the antistress effect of SWLT, the effect of the coexistence of Asp with caffeine, EGCG, Thea, and Arg was investigated. Even if Asp coexisted, adrenal hypertrophy was suppressed if the CE/TA ratio in the drinking water was 0.5 or less (Figure 4). The antagonistic effect of Asp on Thea and Arg was found to be weaker than that of caffeine and EGCG.

It has been confirmed that similar results can be obtained at different concentrations if the CE/TA ratio is the same [9]. However, it was newly revealed that Asn and Asp act antagonistically to the antistress effect of Thea (Figure 2). Therefore, to clarify the role of Asp in the antistress effect of SWLT, the effect of the coexistence of Asp with caffeine, EGCG, Thea, and Arg was investigated. Even if Asp coexisted, adrenal hypertrophy was suppressed if the CE/TA ratio in the drinking water was 0.5 or less (Figure 4). The antagonistic effect of Asp on Thea and Arg was found to be weaker than that of caffeine and EGCG.

### 2.4. Antidepressant Effects of SWLT

Senescence-accelerated mouse prone 10 (SAMP10) mice are reported to exhibit depression-like behavior [21,22]. These mice were fed SWLT or placebo green tea as drinking water for one month and evaluated by the tail suspension test. A longer immobility time indicates stronger depressive behavior. The immobility behavior was significantly shorter in mice that ingested SWLT compared to the control mice (Figure 5). Depressive behavior tended to be suppressed similarly in mice that ingested placebo tea.

## 3. Discussion

Since it was clarified that the stress-reducing effect of Thea was increased by Arg and suppressed by caffeine and EGCG [9], the stress-reducing effect of green tea has been evaluated using green tea with lowered caffeine, and matcha that is rich in Thea and caffeine [10,11,12,13]. Based on these results, it has been clarified that matcha tea with a CE/TA molar ratio of 2 or less has a stress-reducing effect [13]. As the SWLT used in this experiment contained a large amount of Thea and had a CE/TA ratio of 1.21, a stress-reducing effect was expected. In clinical studies, anxiety, as assessed by the STAI value, tended to be lower with intake of SWLT than the placebo (regular green tea), but the grade of stress measured by sAA or subjective stress was not different between the SWLT and placebo groups. These data suggested that the stress-reducing effect of SWLT was not particularly strong. Therefore, we conducted animal experiments in order to clarify why the stress-reducing effect of SWLT was not as strong as expected, despite very high levels of Thea and Arg.

Thea synthesized in the roots is transferred to the stems and leaves, and is then metabolized to polyphenols—mainly catechins—under light conditions [1]. However, in the dark, this metabolism is suppressed, resulting in Thea accumulation. On the other hand, strong shading causes growth suppression and metabolism changes in tea leaves. Therefore, shading the tea leaves increases the amino acid content [3] but causes a large change in the ratio of each amino acid. The proportion of Thea declined, while the proportions of Arg, Gln, Asn, and Ser increased significantly (Figure 1). The increase in these amino acids is thought to be due to degradation of the soluble protein [4]. Arg has an antistress effect, but Gln and Glu have no effect [9]. Since aspartate is known to act as an excitatory neurotransmitter, similar to Glu [23], we examined whether Asn and Asp, which are abundant in SWLT, affect the antistress activity of Thea. When amino acids were present in the proportions contained in the SWLT, the stress-reducing effect of Thea was not affected in the presence of Arg and Gln, but was canceled by the addition of Asn and Asp (Figure 2). This indicates that the high ratio of other amino acids to Thea, particularly Asn and Asp, is one reason for the low antistress effect of SWLT.

In matcha, a very fine powdered type of green tea, the typical stress response, adrenal hypertrophy, is suppressed in mice if the molar ratio of caffeine and EGCG to Thea and Arg (CE/TA) is 2 or less. In the case of infused green tea with water, the effect of reducing stress was observed in humans when the CE/TA ratio was 0.54 or less [10], but was not observed when the ratio was 0.9 or more [10]. The SWLT used in this study had a CE/TA ratio of 1.12, and when the CE/TA ratio was reduced to 0.45, a stress-reducing effect was observed in mice (Figure 3). From these results, it was clarified that, though the amount of theanine was high in the SWLT, Asp and caffeine were also high, so the expected stress-reduction effect could not be obtained. However, the antidepressant effect of SWLT was significantly higher than in the control. A similar effect was observed in mice that ingested placebo green tea. Tea consumption has been reported to be associated with an antidepressant effect [15]. The amount of caffeine in tea may be an important factor for the antidepressant effect [18]. As long-term stress causes anxiety and depression, the antidepressant effect of SWLT and green tea may be of value in stressful modern societies.

In addition, one of the causes of stress-induced depression may be an imbalance between the excitatory neurotransmitter Glu and the inhibitory neurotransmitter GABA [24]. Changes in Glu and GABA balance cause changes in neuronal excitability, synaptic plasticity, and normal central nervous system function [25]. Therefore, modulating the balance between Glu and GABA may improve stress-induced anxiety and depression.

In the hippocampus of mice that ingested Thea (6 mg/kg) in drinking water for two weeks, the level of Glu was significantly reduced, and, conversely, the level of GABA increased [26]. GABA is synthesized by the decarboxylation of Glu [27]. Altered glutamate decarboxylase (GAD) activity causes an altered GABA level and excitatory/inhibitory balance, but it is not yet known how Thea is involved in regulating GAD activity. However, we recently found that the expression level of the transcription factor neuronal PAS domain protein 4 (*Npas4*) increased in mice that ingested Thea [28]. Since Npas4 regulates the formation and maintenance of inhibitory synapses in response to excitatory synaptic activity, the increased expression of Npas4 by Thea suggests increased GABA release [29]. In addition, Thea inhibits Glu uptake from the Gln receptor, resulting in the inhibition of Glu release [29,30]. That is, when incorporated into the brain, Thea modulates the Glu/GABA balance. On the other hand, caffeine increases the level of Glu in the brain, but not the GABA level [31], and EGCG facilitates Glu release [32]. In addition, EGCG has been reported to suppress overexpression of the GABA pathway and to inhibit GABA by modulating the GABA_A_ receptor [33,34,35]. From these results, Thea, caffeine, and EGCG are considered to influence the Glu/GABA balance in the brain. Furthermore, Asp works as an excitatory neurotransmitter, while Arg reduces physical stress and anxiety through nitric oxide synthesis [36,37]. Arg has also been reported to have an antidepressant effect in rats through increased expression of brain-derived neurotrophic factor [38]. Taken together, these green tea components cause changes in the excitatory/inhibitory balance in the brain due to differences in their composition. That is, SWLT may have antidepressant effects rather than an antistress effect similar to that of regular green tea due to its high levels of caffeine and amino acids. Ketamine, a fast-acting antidepressant, is considered to play a crucial role in the glutamatergic system [39]. Thus, the composition balance of SWLT may have a major impact on the glutamatergic system. A further detailed study is required, including of the interactions between SWLT components.

## 4. Materials and Methods

### 4.1. Effect of SWLT Ingestion on Humans

Tea (*Camellia sinensis* (L.) *Kuntze*) leaves were collected in Shizuoka, Japan in May. About two weeks before harvesting, the leaves were protected from direct sunlight with a shading net [3]. Then, the tea leaves were made into green tea through the usual process. We termed this shaded white leaf tea, or SWLT. One tea bag of SWLT or placebo (normal sencha) tea (3 g of tea in a bag) was steeped in 500 mL of room-temperature water. Tap water was used in this experiment. The participants prepared SWLT or placebo tea every morning and ingested it by the evening. The tea bag was left in the water until the evening. Similarly, after each day’s pharmacy practice, the participants drank these teas. For the measurement of the tea component in the eluate, tea leaves of SWLT (3 g) and placebo tea were steeped in 500 mL of room-temperature water for 3 h and stirred occasionally.

### 4.2. Measurement of the Tea Components by HPLC

The eluates of the SWLT and placebo tea were measured by HPLC, as described previously [9]. In brief, catechins and caffeine in the eluates were measured by HPLC (SCL-10Avp, Shimadzu, Japan; Develosil packed column ODS-HG-5, 150 × 4.6 mm, Nomura Chemical Co. Ltd., Seto, Japan) according to the method of Horie et al. [40]. Catechins and caffeine were measured at 280 nm. Free amino acids in the tea leaves were measured by HPLC as described above, using glycylglycine as an internal standard [41]. Amino acids were detected at an excitation wavelength of 340 nm and at an emission wavelength of 450 nm (RF-535 UV detector, Shimadzu, Japan).

### 4.3. Participants

Forty-eight healthy fifth-year students from the University of Shizuoka participated in the experiment, and were randomly divided into two groups by sex: SWLT (*n* = 24, 11 men and 13 women; average age 22.7 ± 1.1 y) and placebo (*n* = 24, 11 men and 13 women; average age 22.4 ± 0.9 y). They received SWLT or placebo tea bags in sealed envelopes. The participants were assigned to practice outside the university, in a hospital or a dispensing pharmacy, for 11 weeks. The first 10 days of the practice program were analyzed, because these days were assumed to be the most stressful. None of the participants reported having an acute or chronic disease, regular medication intake, or habitual smoking. They were instructed to mainly drink the test tea, and not to consume other theanine- and caffeine-rich beverages such as green tea, coffee, and black tea throughout the experiment. They could drink water freely, but they did not consume alcohol at night. The study was conducted in accordance with the Declaration of Helsinki and Ethical Guidelines for Medical and Health Research Involving Human Subjects (Public Notice of the Ministry of Education, Culture, Sports, Science and Technology and the Ministry of Health, Labour and Welfare, 2008). The study protocol was approved by the Ethics Committee of the University of Shizuoka (No. 29-57). All the participants received verbal and written information about the study and signed an informed consent form before entering the study. This study was registered at the University Hospital Medical Information Network (UMIN) (registration ID no. UMIN 000031778). The study period was April to May 2018.

### 4.4. Procedure

This study was a group comparison design and the participants were randomly assigned to the SWLT or placebo tea groups. The participants did not know whether they were consuming SWLT or placebo tea, because they had no information about the SWLT’s aroma and taste. The intake of SWLT or placebo tea was from one week prior to pharmacy practice and continued for 10 days into the practice period, for a total of 17 days. All the collected data were used for the analysis. To assess the anxiety of the participants, the state-trait anxiety inventory (STAI) test (Japanese STAI Form X-1, Sankyobo, Kyoto, Japan) was carried out before and after their pharmacy practice. A questionnaire that included feedback on physical condition, subjective stress, and sense of achievement was assigned for 10 days after each day’s practice. The physical condition of participants was assessed on an ordinal scale (5, very good; 4, good; 3, normal; 2, somewhat poor; 1, bad). Subjective stress was evaluated using visual analogue scales (VAS: 0–10), from very relaxed to highly stressed. Sense of achievement was assigned an ordinal scale (5, complete; 4, better; 3, a little better; 2, a little worse; 1, much worse) [8]. Sleep hours were also recorded.

### 4.5. Measurement of Salivary Amylase Activity (sAA)

An oral cavity enzyme, sAA, is generally used a marker of stress. To assess the physiological stress response, sAA was measured using a colorimetric system (Nipro Co., Osaka, Japan), as described previously [42]. Saliva was collected twice a day, in the morning after waking up (prepractice) and in the evening after pharmacy practice (postpractice), for 10 days during the practice. To establish a baseline of sAA before the pharmacy practice, the sAA of participants was measured every morning and evening for seven days during routine daily life at the university. The measurement was carried out before pharmacy practice. Prior to sampling, participants washed their mouths with water. After saliva was collected for 30 s using a sampling tip, each participant measured their own sAA immediately every morning and evening for 17 days.

### 4.6. Animals and the Stress Experiment

Male ddY mice (Slc: ddY, four weeks old) were purchased from Japan SLC Co. Ltd. (Shizuoka, Japan) and kept under conventional conditions in a temperature and humidity-controlled environment with a 12/12 h light/dark cycle (light period, 8:00 a.m.–8:00 p.m.; temperature, 23 ± 1 °C; relative humidity, 55 ± 5%). The four-week-old mice were housed in groups of four in a cage for five days to allow them to adapt to cohabitation. The mice were fed a normal diet (CE-2; Clea Co. Ltd., Tokyo, Japan) and water ad libitum. All experimental protocols were approved by the University of Shizuoka Laboratory Animal Care Advisory Committee (approval No. 195241) and were in accordance with the guidelines of the U.S. National Institute of Health for the Care and Use of Laboratory Animals. The mice were then divided into two groups: the confrontational group and group housing, according to a previously described method [7]. To apply psychosocial stress to the mice, confrontational housing was established in a standard polycarbonate cage (16 × 27 cm) that was divided into two identical subunits by a stainless-steel partition, as described previously [7]. In brief, two male mice were housed in a partitioned cage for six days (single housing) to establish territorial consciousness. Then, the partition was removed to expose the mice to confrontational stress for 24 h (confrontational housing). Each cage was placed in a Styrofoam box (width 30 cm, length 40 cm, height 15 cm) to avoid visual social contact between cages. At the end of the 24 h of confrontational housing, the mice were sacrificed and the adrenal glands were weighed.

### 4.7. Ingestion of Tea Components by Mice

The effect of the tea components on the stress response was examined in mice (four mice/group, *n* = 84). Two mice were housed in a partitioned cage for six days (single housing). Then, the partition was removed and subsequently the two mice cohabited the same cage (confrontational housing). The mice consumed tea components in water ad libitum for seven days (single housing for six days and confrontational housing for one day). Mouse body weight was measured on the last day of the experiment. Group housing mice were used as a control. In SWLT, Gln, Asn and Asp were abundant compared to the placebo tea. Based on the amino acid concentration in the SWLT eluate (Table 1), the relationships among amino acids were examined. Each amino acid concentration was as follows: Thea 140 mg/L, Arg 70 mg/L, Gln 52 mg/L, Asn 34 mg/L, and Asp 34 mg/L. The stress-reducing effect on adrenal hypertrophy was compared among these groups as follows: Group 1, no amino acids; Group 2, Thea; Group 3, Thea, Arg, and Gln; Group 4, Thea, Arg, Gln, and Asn; Group 5, Thea, Arg, Gln, Asn, and Asp.

Next, the effects of caffeine and EGCG on the antistress effect of Thea and Arg were examined in mice under confrontational housing conditions. These mice were separated into four groups as follows: Group 1; control. Group 2; mice ingested water containing the same concentrations of caffeine (103 µM), EGCG (33 µM), Thea (80 µM) and Arg (40 µM) as SWLT (CE/TA = 1.12). Group 3; mice ingested water containing caffeine (36 µM), EGCG (15 µM), Thea (80 µM), and Arg (40 µM). The CE/TA ratio of this composition was 0.45. Group 4; mice ingested water containing the same concentrations of caffeine (58 µM), EGCG (29 µM), Thea (16 µM) and Arg (3 µM) as placebo tea (CE/TA = 4.47).

Furthermore, Asp was added, and the antagonistic effect of caffeine and EGCG against the antistress effect of Thea and Arg was investigated in mice under confrontational housing conditions. These mice were separated into seven groups, as follows: Group 1; control. Group 2; mice ingested water containing the same concentrations of caffeine (30 µM), EGCG (30 µM), Thea (80 µM) and Arg (40 µM) (CE/TA = 0.5). Group 3; Group 2 + Asp (90 µM). Group 4: mice ingested water containing caffeine (60 µM), EGCG (30 µM), Thea (80 µM), and Arg (40 µM) (CE/TA = 0.75). Group 5; Group 4 + Asp (90 µM). Group 6; mice ingested water containing the same concentrations of caffeine (60 µM), EGCG (60 µM), Thea (80 µM) and Arg (40 µM) (CE/TA = 1.0). Group 7; Group 6 + Asp (90 µM). The tea components used were as follows: L-theanine and EGCG (Taiyo Kagaku Co. Ltd., Yokkaichi, Japan), caffeine, Arg, Gln, Asp, and Asn (Wako Pure Chemical Co. Ltd., Osaka, Japan).

### 4.8. Tail Suspension Test

Male SAMP10/TaIdrSlc mice were purchased from Japan SLC (Shizuoka, Japan). The four-week-old mice were housed in a group of six in a cage. The mice were fed a normal diet (CE-2; Clea Co. Ltd., Tokyo, Japan) and water containing tea components of SWLT or placebo green tea ad libitum. The concentration of each component of SWLT or placebo green tea was matched to the intake per human body weight per day. These mice ingested SWLT or placebo green tea for one month. The tea components used, other than those mentioned in Section 4.7, were as follows: EGC and EC (Taiyo Kagaku Co. Ltd., Yokkaichi, Japan), Glu and Ser (Wako Pure Chemical Co. Ltd., Osaka, Japan). The control mice drank tap water. To investigate behavioral depression, the mice were individually suspended by their tails at a height of 30 cm using a clip for tail suspension (MSC2007; YTS Yamashita-Giken, Tokushima, Japan). The immobility behavior was observed for 15 min, as described previously [7].

### 4.9. Statistical Analysis

These results are expressed as the mean ± SEM. The influence of stress on sAA was evaluated using a one-way analysis of variance (ANOVA), followed by a Wilcoxon/Kruskal‒Wallis post hoc test or Fisher’s least significant difference test for multiple comparisons. In each analysis, a *p*-value < 0.05 was considered to be statistically significant.

## 5. Conclusions

The antistress effect of SWLY was compared with common green tea in a clinical study. However, the stress-reducing effect of SWLT in students was not high. Next, the effects on stress of caffeine and amino acids, which are abundant in SWLT, were examined in animal experiments. Furthermore, we observed the depression-like behavior of mice that ingested SWLT or common green tea. The results showed that SWLT, which has higher levels of caffeine and amino acids such as Thea, Arg, Gln, Asn, and Asp than common green tea, may have an antidepressant effect.

This study had some limitations. The participants were solely young students, so it is necessary to investigate participants of different ages. The students were tested under conditions of mild stress, but different stress conditions should be considered in the future.

## Figures and Tables

**Figure 1 molecules-25-03550-f001:**
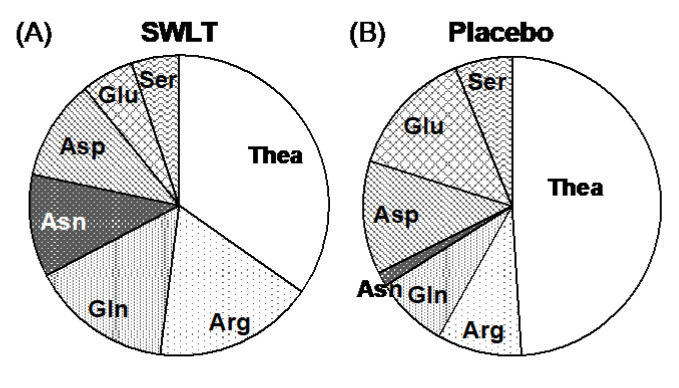
Molar ratio of free amino acids in SWLT and placebo green tea. (**A**) SWLT, (**B**) Placebo green tea.

**Figure 2 molecules-25-03550-f002:**
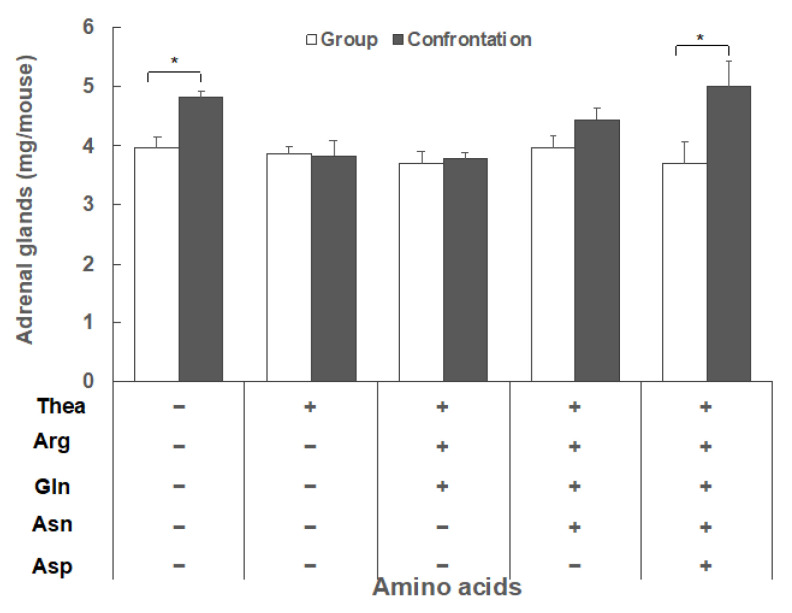
Effect of amino acid composition on the stress response in mice. Each amino acid concentration is the same in SWLT, as follows: Thea 140 mg/L, Arg 70 mg/L, Gln 52 mg/L, Asn 34 mg/L, and Asp 34 mg/L. Two mice were housed in a partitioned cage for six days (single housing). Then, the partition was removed and subsequently the two mice cohabited the same cage for one day (confrontational housing). Group housing mice were housed in groups of four. These mice ingested water containing amino acids ad libitum. Data are shown as mean ± SEM (*n* = 4–8, * *p* < 0.05).

**Figure 3 molecules-25-03550-f003:**
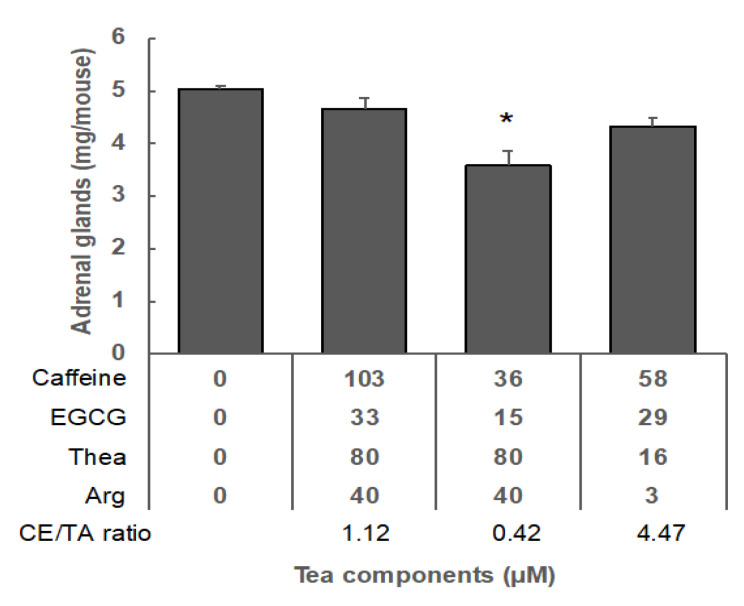
Effect of tea components on the stress response in mice under confrontational housing conditions. After single housing for six days, the mice were housed confrontationally for one day. These mice were separated into four groups, as follows: Group 1; control. Group 2; mice ingested water containing the same concentrations of caffeine (103 µM), EGCG (33 µM), Thea (80 µM) and Arg (40 µM) as SWLT (CE/TA = 1.12). Group 3; mice ingested water containing caffeine (36 µM), EGCG (15 µM), Thea (80 µM), and Arg (40 µM). The CE/TA ratio of this composition was 0.45. Group 4; mice ingested water containing the same concentrations of caffeine (58 µM), EGCG (29 µM), Thea (16 µM), and Arg (3 µM) as in placebo green tea (CE/TA = 4.47). Data are shown as mean ± SEM (*n* = 4, * *p* < 0.05).

**Figure 4 molecules-25-03550-f004:**
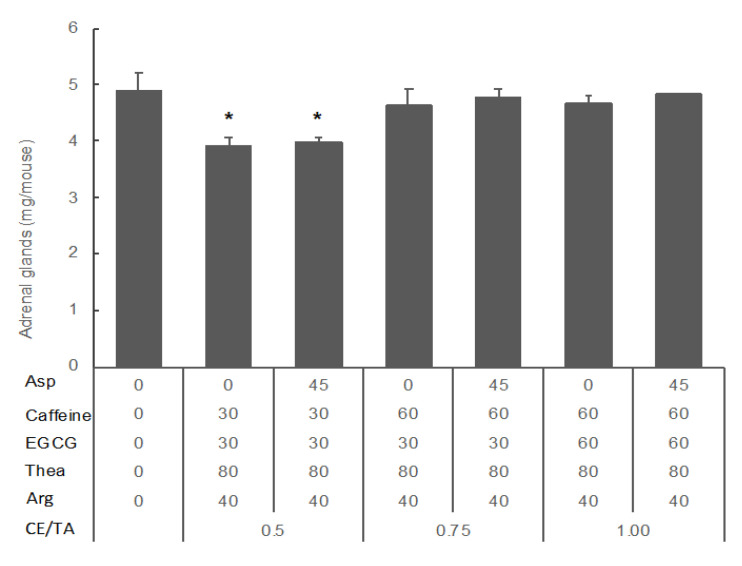
Effect of tea components on the stress response in mice under confrontational housing conditions. After single housing for six days, the mice were housed confrontationally for one day. These mice ingested water containing Asp (45 µM), caffeine (30 or 60 µM), EGCG (30 or 60 µM), Thea (80 µM), and Arg (40 µM). Data are shown as mean ± SEM (*n* = 3–4, * *p* < 0.05).

**Figure 5 molecules-25-03550-f005:**
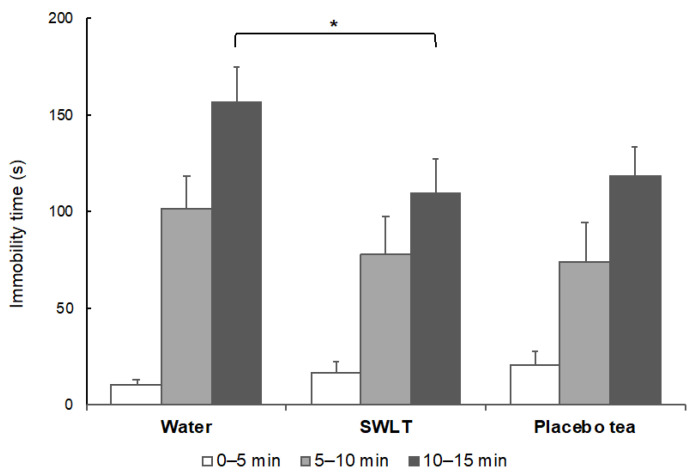
Effect of SWLT ingestion on depression-like behavior of SAMP10 mice. These mice ingested SWLT or placebo green tea as drinking water for one month and were evaluated using the tail suspension test. The control mice drank water. Data are shown as mean ± SEM (*n* = 5–6, * *p* < 0.05).

**Table 1 molecules-25-03550-t001:** The contents of caffeine, catechins, and amino acids in the eluate of SWLT and placebo tea.

Tea	Caffeine (mg/L)	Catechins (mg/L)							
		EGCG	EGC	ECG	EC	GC	CG	(+)C	
**SWLT**	209.8	150.4	135.2	24.6	41.0	5.0	2.8	3.4	
**Placebo**	112.0	134.2	229.0	21.0	46.6	13.6	4.6	2.0	
**Tea**	**Free amino acids (mg/L)**								
	**Thea**	**Arg**	**Gln**	**Asn**	**Asp**	**Glu**	**Ser**	**GABA**	**Total**
**SWLT**	140.2	69.9	51.7	33.8	33.5	19.3	12.6	0	361.0
**Placebo**	28.8	5.4	3.9	0.7	5.5	6.9	2.2	0	53.5

Shaded white leaf tea (SWLT) and placebo green tea (3 g) were steeped in 500 mL of room-temperature water for 3 h. EGCG, (−)-epigallocatechin gallate; EGC, (−)-epigallocatechin; ECG, (−)-epicatechin gallate; EC, (−)-epicatechin; CG, (−)-catechin gallate; (+) C, (+)-catechin; Thea, theanine; Arg, arginine; Gln, glutamine; Asn, asparagine; Asp, aspartic acid; Glu, glutamic acid; Ser, serine; GABA, γ-aminobutyric acid.

**Table 2 molecules-25-03550-t002:** Effect of SWLT intake on each test item.

Test item	Practice		SWLT	Placebo tea	*p*
**sAA**	University	Pre-practice	28 ± 4	24 ± 3	0.484
		Post-practice	45 ± 3	47 ± 3	0.645
	Pharmacy	Pre-practice	28 ± 4	22 ± 3	0.231
		Post-practice	46 ± 7	34 ± 5	0.292
**STAI value**		Before the pharmacy practice	45 ± 3	47 ± 3	0.645
		After the pharmacy practice	40 ± 3	46 ± 3	0.065
**Physical condition** (score 1–5)		University	2.5 ± 0.2	2.7 ± 0.1	0.482
		Pharmacy	2.6 ± 0.1	2.6 ± 0.1	0.828
**Subjective stress** (VAS: 0–10)		University	4.2 ± 0.3	4.6 ± 0.3	0.395
		Pharmacy	4.6 ± 0.4	4.6 ± 0.3	0.765
**Achievement emotion** (score 1–5)		Pharmacy	2.4 ± 0.2	2.5 ± 0.2	0.567
**Sleeping time** (h)		University	6.5 ± 0.1	6.4 ± 0.2	0.536
		Pharmacy	6.4 ± 0.2	6.5 ± 0.2	0.508

These data represent the mean ± standard error of the mean (SEM) (SWLT *n* = 24, placebo *n* = 24).

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
