# Peer review of "Antidepressant Effect of Shaded White Leaf Tea Containing High Levels of Caffeine and Amino Acids"

_molecules, 2020, doi:10.3390/molecules25153550_

Round 1

Reviewer 1 Report

The authors evaluated the antidepressant and anti-stress effects of shaded white leaf tea with high levels of caffeine and amino acids, compared with regular green tea as placebo. However, the chemical compositions of shaded white leaf tea were quite different from green tea (placebo), in terms of the levels of caffeine, catechins and amino acids and some undetermined compounds, which was difficult to explain the reason for the antidepressant activity. It was a bit abnormal that the total concentration of amino acids in white leaf was even higher than that of catechin compounds. Although Thea, Arg, caffeine and EGCG were used for constructing artificial formula for investigating the effect of composition ratio in animal studies, the effect of concentrations were not ruled out yet. Besides, English expression need improvement, and many places are very difficult to understand. In my opinion, the manuscript is not suitable for publication in Molecules.

Author Response

Response to reviewer 1

The authors evaluated the antidepressant and anti-stress effects of shaded white leaf tea with high levels of caffeine and amino acids, compared with regular green tea as placebo. However, the chemical compositions of shaded white leaf tea were quite different from green tea (placebo), in terms of the levels of caffeine, catechins and amino acids and some undetermined compounds, which was difficult to explain the reason for the antidepressant activity. It was a bit abnormal that the total concentration of amino acids in white leaf was even higher than that of catechin compounds. Although Thea, Arg, caffeine and EGCG were used for constructing artificial formula for investigating the effect of composition ratio in animal studies, the effect of concentrations were not ruled out yet. Besides, English expression need improvement, and many places are very difficult to understand. In my opinion, the manuscript is not suitable for publication in Molecules.

Thanks for commenting.

We have already found that the relative ratio of Thea, Arg, caffeine, and EGCG, but not the absolute amount, is important (Unno et al 2015). Therefore, we added as follows in the result of Fig. 4:

It has been confirmed that similar results can be obtained at different concentrations if the CE/TA ratio is the same [9]. However, …

This manuscript had been proofread in English by MDPI, but since it was revised and corrected this time, it was checked again in English by MDPI.

Reviewer 2 Report

Lines 2 to 4: The title of your manuscript is too long, I suggest it to be to rewritten.

Lines 22 to 23 : I suggest authors to write : These leaves are called “shaded white leaf tea”, or SWLT. In the eluate of SWLT, the amount of amino acids is about 5–7 times higher than that of common green tea (SWLT: 361 mg/L, Placebo : 53.5 mg/L).

Lines 187 to 188, add reference.

Line 242 : Tea (Camellia sinensis (L.) Kuntze) leaves were collected in Shizuoka, Japan. I suggest Authors to specify the collecting period.

According to the data in table 1, you wrote in the abstract (L. 23) that "arginine is the second most abundant compound of the amino acid in tea". What about Glutamine, I suggest you to make a one factor ANOVA for each line of this table, in order to show if there is a significant difference between the data of a same line.

Line 353: Could I know the surface (m2) of the cages? it would be appropriate for the dimensions of experimental cages to be specified.

Line 414: Reference 14 must be written correctly.

Author Response

Response to reviewer 2

Lines 2 to 4: The title of your manuscript is too long, I suggest it to be to rewritten.

Thank you very much for reviewing our manuscript. The title was shortened as follows: Antidepressant Effect of Shaded White Leaf Tea Containing High Levels of Caffeine and Amino Acids

Lines 22 to 23 : I suggest authors to write : These leaves are called “shaded white leaf tea”, or SWLT. In the eluate of SWLT, the amount of amino acids is about 5–7 times higher than that of common green tea (SWLT: 361 mg/L, Placebo : 53.5 mg/L).

To revise according to the suggestion and within 200 words, we revised as follows: the amount of amino acid (361 mg/L) was significantly higher than that in regular tea (53.5 mg/L)  

Lines 187 to 188, add reference.

Reference was newly added (line 202).

Line 242 : Tea (Camellia sinensis (L.) Kuntze) leaves were collected in Shizuoka, Japan. I suggest Authors to specify the collecting period.

Since the leaves were collected in May, the collection period "in May" was added to the text.

According to the data in table 1, you wrote in the abstract (L. 23) that "arginine is the second most abundant compound of the amino acid in tea". What about Glutamine, I suggest you to make a one factor ANOVA for each line of this table, in order to show if there is a significant difference between the data of a same line.

The reviewers recommended making a one-way analysis of variance for each row in Table 1 to see if there was a significant difference between Arg and Gln. However, we focused on Arg because it has the similar anti-stress effect as Thea, but not because it is the second most abundant. On the other hand, Gln has no anti-stress effect. Therefore, I don't think there is a need to compare quantitative differences.

Line 353: Could I know the surface (m2) of the cages? it would be appropriate for the dimensions of experimental cages to be specified.

We added the size of cage in the section 4.6.

Line 414: Reference 14 must be written correctly.

This is the address of the report on the Japanese Ministry of Health, Labor and Welfare website.

Reviewer 3 Report

Reviewer comment:

- The manuscript entitled ,,Shaded White Leaf Tea with High Levels of Caffeine and Amino Acids May Have an Antidepressant Rather Than an Anti-Stress Effect'' provides interesting data on this topic.

This record is a continuation of work on the properties of tea grown and produced in different ways. This is a nicely written contribution based on clearly described experiments. The topic of this study is interesting and in my opinion it could be interesting for a reasonable number of scientists. Further in vitro, in vivo and clinical studies are needed to better investigate the processes described and to learn about side effects of Shaded White Leaf Tea.

I have only a small comment:

- Figures: under figures 2-5 no description ± SEM

- page correction - References:

- line 387: please insert ;  instead :

5 Unno, K.; Fujitani, K.; Takamori, N.; Takabayashi, F.: Maeda, K.; Miyazaki, H.; Tanida, N.; Iguchi, K.; Shimoi, K.; Hoshino, M. Theanine intake improves the shortened lifespan, cognitive dysfunction and behavioural depression that are induced by chronic psychosocial stress in mice. Free Radic Res 2011, 45, 966–974.

After a slight correction, I suggest acceptance of this paper.

Author Response

Response to reviewer 3

The manuscript entitled, Shaded White Leaf Tea with High Levels of Caffeine and Amino Acids May Have an Antidepressant Rather Than an Anti-Stress Effect'' provides interesting data on this topic.

This record is a continuation of work on the properties of tea grown and produced in different ways. This is a nicely written contribution based on clearly described experiments. The topic of this study is interesting and in my opinion it could be interesting for a reasonable number of scientists. Further in vitro, in vivo and clinical studies are needed to better investigate the processes described and to learn about side effects of Shaded White Leaf Tea.

I have only a small comment:

- Figures: under figures 2-5 no description ± SEM

Thank you for reviewing our manuscript. We added in Fig. 2-5 as follows, “Data are shown as mean ± SEM”.

- page correction - References:

- line 387: please insert ;  instead :

5 Unno, K.; Fujitani, K.; Takamori, N.; Takabayashi, F.: Maeda, K.; Miyazaki, H.; Tanida, N.; Iguchi, K.; Shimoi, K.; Hoshino, M. Theanine intake improves the shortened lifespan, cognitive dysfunction and behavioural depression that are induced by chronic psychosocial stress in mice. Free Radic Res 2011, 45, 966–974.

Thank you for finding the mistype.

Reviewer 4 Report

In the present paper, the Authors first examined whether "shaded white leaf 50 tea” (SWLT) can reduce stress, because of the high amounts of Theanine (Thea) and Arginine (Arg). However, They found that the stress-reducing effect of SWLT in students was about the same as placebo green tea. Therefore, the Authors investigated the reasons for this unexpected result for SWLT using an animal psychosocial stress model, and based on this data, they examined the antidepressant effect of SWLT.
Overall, I found this study very interesting, timely, innovative and well conducted. However, I have several concerns on it that should be addressed prior publication and these are outlined below:
1) I believe that, in the introduction, a brief note on depression (and differences with bipolar disorder) should be given with appropriate references (see Han KM et al. Prog Neuropsychopharmacol Biol Psychiatry 2019;91:20-27 and Ventriglio et al. Int Rev Psychiatry. 2020 May 21:1-6). Moreover a brief note on suicidal ideation would be useful (see Orsolini et al. Psychiatry Investig 2020;17:207-221 and De Berardis et al. Early Interv Psychiatry. 2020 Jun;14(3):336-342)
2) Novel antidepressant targets include the glutamatergic system that may be indirectly involved in the effect of SWLT. Please comment this with appropriate references (I suggest De Berardis et al. Curr Top Med Chem. 2020;20(7):554-584 and Int J Mol Sci. 2018 Sep 23;19(10):2888).
3) How the forty-eight healthy 5th-year students from the University of Shizuoka were selected and enrolled) How the participants were assessed concerning acute or chronic disease, regular medication intake, or habitual smoking?
4) Was any recompense provided for study' participation?
5) The Conclusion paragraph is too short (only two lines). Please, summarize here the main research' findings and the future studies directions. Moreover, please discuss study' limitations.

6) Please carefully check the English Language in some points.

Author Response

Response to reviewer 4

In the present paper, the Authors first examined whether "shaded white leaf 50 tea” (SWLT) can reduce stress, because of the high amounts of Theanine (Thea) and Arginine (Arg). However, They found that the stress-reducing effect of SWLT in students was about the same as placebo green tea. Therefore, the Authors investigated the reasons for this unexpected result for SWLT using an animal psychosocial stress model, and based on this data, they examined the antidepressant effect of SWLT.

Overall, I found this study very interesting, timely, innovative and well conducted. However, I have several concerns on it that should be addressed prior publication and these are outlined below:

  • I believe that, in the introduction, a brief note on depression (and differences with bipolar disorder) should be given with appropriate references (see Han KM et al. Prog Neuropsychopharmacol Biol Psychiatry 2019;91:20-27 and Ventriglio et al. Int Rev Psychiatry. 2020 May 21:1-6). Moreover a brief note on suicidal ideation would be useful (see Orsolini et al. Psychiatry Investig 2020;17:207-221 and De Berardis et al. Early Interv Psychiatry. 2020 Jun;14(3):336-342)

Thank you for your review and your very valuable suggestions, including some papers on depression. As you pointed out, we briefly explained depression in the introduction. The proposed paper was not cited directly, but we further examined based on these papers. As a result, in addition with a brief note on depression, we also explained the relationship between depression and green tea.

2) Novel antidepressant targets include the glutamatergic system that may be indirectly involved in the effect of SWLT. Please comment this with appropriate references (I suggest De Berardis et al. Curr Top Med Chem. 2020;20(7):554-584 and Int J Mol Sci. 2018 Sep 23;19(10):2888).

We commented this in the discussion using the suggested reference.

3) How the forty-eight healthy 5th-year students from the University of Shizuoka were selected and enrolled) How the participants were assessed concerning acute or chronic disease, regular medication intake, or habitual smoking?

Participants themselves answered the questionnaire for these questions.

4) Was any recompense provided for study' participation?

Yes, we thanked the participants for their supportive daily measurements and records and paid a reasonable amount as a prepaid card.

5) The Conclusion paragraph is too short (only two lines). Please, summarize here the main research' findings and the future studies directions. Moreover, please discuss study' limitations.

Thank you for your suggestion. The conclusion was revised, and we summarized the main research and findings in it. In addition, we discussed the study limitations.

6) Please carefully check the English Language in some points.

This manuscript had been proofread in English by MDPI, but since it was revised and corrected this time, it was checked again in English.